# Reliability and Correlation of Different Devices for the Evaluation of Primary Implant Stability: An In Vitro Study

**DOI:** 10.3390/ma14195537

**Published:** 2021-09-24

**Authors:** Perry Raz, Haya Meir, Shifra Levartovsky, Maia Peleg, Alon Sebaoun, Ilan Beitlitum

**Affiliations:** 1Department of Periodontology and Dental Implants, The Maurice and Gabriela Goldschleger School of Dental Medicine, The Sackler Faculty of Medicine, Tel Aviv University, Tel Aviv 6997801, Israel; hayameir@012.net.il (H.M.); alon.sebaoun@gmail.com (A.S.); beilan1612@gmail.com (I.B.); 2Department of Oral Rehabilitation, The Maurice and Gabriela Goldschleger School of Dental Medicine, The Sackler Faculty of Medicine, Tel Aviv University, Tel Aviv 6997801, Israel; shifralevartov@gmail.com; 3The Maurice and Gabriela Goldschleger School of Dental Medicine, The Sackler Faculty of Medicine, Tel Aviv University, Tel Aviv 6997801, Israel; pelegmaia@gmail.com

**Keywords:** primary stability, implant stability, ISQ, RFA, insertion torque, implant design, bone density

## Abstract

Our aim was to analyze the correlation between the IT evaluated by a surgical motor and the primary implant stability (ISQ) measured by two RFA devices, Osstell and Penguin, in an in vitro model. This study examines the effect of bone type (soft or dense), implant length (13 mm or 8 mm), and implant design (CC: conical connection; IH: internal hexagon), on this correlation. Ninety-six implants were inserted using a surgical motor (IT) into two types of synthetic foam blocks. Initial measurements for both the peak IT and ISQ were recorded at the point when implant insertion was stopped by the surgical motor, and the final measurements were recorded when the implant was completely inserted into the synthetic blocks using only the RFA devices. Our null hypothesis was that there is a good correlation between the devices, independent of the implant length, design, or bone type. We found a positive, significant correlation between the IT, and the Osstell and Penguin devices. Implant length and bone type did not affect this correlation. The correlation between the devices in the CC design was maintained; however, in the IH design it was maintained only between the RFA devices. We concluded that there is a high positive correlation between the IT and ISQ from a mechanical perspective, which was not affected by bone type or implant length but was affected by the implant design.

## 1. Introduction

Implant primary stability is an important parameter in optimal osseointegration and is a critical prerequisite factor for immediate or early loading [1,2,3]. Primary implant stability is affected by various factors, such as the implant’s geometry, bone quality and quantity (mainly cortical bone thickness), and the surgical drilling technique used [3,4,5,6,7]. There are also special surgical implant preparation site protocols that are adjusted to the bone type in order to modify and improve the implant’s primary stability [7,8].

The primary implant stability can be evaluated by the insertion torque (IT), which is measured by the implantation equipment itself without the need for any additional measuring devices [9]. Noninvasive devices based on resonance frequency analysis (RFA) allow clinicians to measure implant stability at different time points in the process of osseointegration to assist in functional loading decision making. The RFA can be measured by the Osstell device (Integration Diagnostics AB, Göteborg, Sweden), which measures the stiffness and deflection of the implant–bone complex [10]. RFA is dependent upon the design of the transducer itself; the stiffness of the implant fixture and its interface with the tissues and surrounding bone; and the total effective length above the marginal bone level [11]. The frequency values obtained by the RFA devices are automatically translated into an index called the implant stability quotient (ISQ), which ranges from 0–100. The Osstell system is considered to have an almost perfect repeatability and reproducibility outcome [3]. The Penguin (Integration Diagnostics, Sweden) is a similar device, with the exception that it is electronic, as opposed to the Osstell, which is a magnetic detection device [12]. Rittel and his colleague, in their laboratory study, claimed that the sensitivity of RFA to changes in the mechanical properties of periprosthetic tissue seems relatively weak and that RFA might be better adapted to bone-healing estimations [13].

The correlation between the IT and RFA has been investigated in numerous studies but is still unclear. Some authors claim that the two parameters are in a direct relationship [14,15], while others have demonstrated that there are no statistically significant correlations between the two. These discrepancies concerning the clinical significance of the IT and RFA values could lead to miscommunication between clinicians regarding the appropriate implant loading time point [16,17,18].

Our hypothesis was that the different devices could provide similar values for primary implant stability for different bone types, implant lengths, and implant designs, and we expected to find a correlation between the IT and ISQ measurements conducted by the Penguin and Osstell devices.

The objective of this study was to evaluate the correlation between the IT and ISQ measurements for these two RFA devices in an in vitro model and examine the influence of bone type, implant length, and implant design on this correlation.

## 2. Materials and Methods

Ninety-six implants (MIS, implant technologies) were inserted in an artificial bone material made of synthetic polyurethane foam blocks with dimensions of 120 mm × 170 mm × 42 mm (Sawbones, Malmö, Sweden) and different cortical thicknesses and trabecular densities. The density of the soft bone block (#10) was 0.16 g/cc, and it was laminated on one side with 1.5 mm of dense cortical bone (#50) with a density of 0.8 g/cc. The dense bone blocks (#40) were characterized by a density of 0.64 g/cc and were laminated on one side with 2 mm of cortical bone (#50). The additional mechanical properties of the bone blocks used in the study are described in the Sawbones catalog [6].

Two implant designs, a tapered internal hexagon (IH) (Seven^®^ new design MIS^®^ Implants Technology Ltd., Misgav, Israel) and a less tapered conical connection (CC) (C1^®^ MIS^®^) Implants Technology Ltd., Misgav, Israel), of two different lengths (13 mm and 8 mm) were used. There were 8 experimental groups consisting of 12 implants in each group (Table 1). The implants were inserted at constant distances of 30 mm from each other across the block according to the manufacturers’ protocols. The implant peak IT was evaluated using a surgical motor with torque control (N/cm) (Implanted, W&H, Burnoose, Austria). The peak IT was recorded from the surgical device display.

Insertion to the full length of the implant was carried out in two steps, since only partial insertion was feasible using the motor device (defined as “initial”). Full-length implant insertion to the bone level was then completed manually using a hand ratchet (defined as “final”). The stability of each implant was measured by a blinded examiner using the Osstell (Integration Diagnostics AB, Göteborg, Sweden) ISQ and the Penguin (Penguin Integration Diagnostics, Göteborg, Sweden) RFA devices after the transducer (Smartpeg) was screwed to the implant. Three repeated measurements from three different angles were recorded by each RFA device for each implant at partial and full implant length insertions.

The characteristics of the two implant designs are as follows: The IH is characterized by a tapered design, and the inter-thread distance is 2 mm, while the CC is less tapered, and its inter-thread distance is 1.5 mm. The CC has two spiral channels in its apex, while the IH has three. The CC has a conical connection, while the IH has an internal hexagonal connection. The threads in the IH are deeper than those in the CC.

Both implants share some design features, including threads that condense at the neck of the implant and cutting threads at their apex. The apexes of both implants are dome-shaped, and both have platform switching microgaps (Figure 1).

Statistical analysis: The significance of the differences in ISQ, RFA, and IT among the groups was assessed by the Kruskal–Wallis test, followed by the Bonferroni corrected Mann–Whitney *U*-test for pairwise comparisons. The correlation of IT with the ISQ and RFA was assessed using the Spearman rho correlation coefficient. The level of significance was set at α = 0.05.

## 3. Results

The mean values for implant stability from the three measurements performed by each device for all tested groups are presented in Table 2. The mean IT for all samples was 41.7 ± 5.44 (N/cm). Higher values for the Osstell and Penguin devices were observed in the final measurements, when the entire implant length that was inserted ranged from 63.95 to 65.08 (ISQ), as compared to the initial measurements, when the implant was partially inserted to a length ranging from 56.87 to 55.07. Lower standard deviations represented more reproducible measurements, as was found for the IT of the machine and for the Osstell and Penguin devices in the initial measurement. Higher standard deviations were observed for the final measurements of both the Osstell and Penguin devices. Three implants were unavailable for measurements due to technical problems—namely, due to a loss of primary stability or an improper insertion depth.

Table 3 presents the mean values of the implant stability of the three measurements performed by each device according to the different groups and is sorted according to the similar parameters of soft or dense bone, the two different implant designs (CC or IH), and short or long implants. The average percentages of the inserted lengths are also presented. Higher values were measured for each implant setting at the final measurement versus the initial measurement when the implant was partially inserted. Despite the implants being inserted deeper into the soft bone than into the dense bone (by more than 80%), the primary stability in the dense bone was higher for both the Osstell and Penguin devices, except for the CC implant inserted at a 13 mm length into dense bone, where the initial measurements were lower versus the same implant in soft bone. When adjusting the bone type and the length, the CC implants were always more stable than the IH implants, except for the CC implant inserted at a 13 mm length into dense bone in the initial measurements. The reason for this may be related to the difficulty in fully inserting the implant, as can be inferred by the short insertion length (61%). A higher stability was observed in the dense bone than in the soft bone for the same implant design and length. A higher stability was also observed for the long implants versus the short implants with the same bone type and implant design.

The distribution of the IT was normal, but the Osstell and Penguin measurements were not normally distributed. Therefore, to analyze the correlation we used the nonparametric correlation of Spearman’s rho. We found a significant positive correlation between the IT and the Osstell and Penguin RFA devices in the initial and final measurements.

Highly positive and significant correlations (with a measurement close to 1) were found between the Osstell and Penguin devices at the initial and final measurements (Table 4).

The correlation between the tested devices was examined in the different bone types to understand the influence of the bone type on the primary implant stability. It was observed that the IT and initial measurements for the Penguin and the Osstell devices were not correlated in dense bone. However, in soft bone, these correlations were positive and significant. The final measurements and IT determined via the Osstell and Penguin devices were positive and significant in both dense and soft bone. The bone type did not affect the high positive correlation between the initial and final Osstell and Penguin values (Table 5).

The correlation between the tested devices was examined for implant lengths of 8 or 13 mm. We found that the correlation between them was high, positive, and significant (Table 6). The implant length did not affect the correlation between the primary stability measurements for the different RFA devices.

The correlation between the tested devices was examined in the different implant design types of the CC and IH. In the CC implant design, the correlation between the devices was maintained, but in the IH implant design the correlation was high and positive only between the Osstell and the Penguin in the initial and final measurements. In the IH implant design, no correlation was observed between the IT and the Osstell device or between the IT and the Penguin device (Table 7).

No differences between the intergroup bone blocks were observed (data are not presented).

## 4. Discussion

Primary stability is an essential parameter for immediate implant installation and early loading. The purpose of our research was to evaluate the correlation between the primary implant stability recorded by the IT and ISQ measurements of two implant designs, and of the different implant lengths in two types of soft and dense bone block models.

As expected, our study showed higher ISQ measurements for both the Osstell and Penguin devices in the final measurements when the entire implant length was inserted versus the initial measurements when the implant was only partially inserted. Reproducible measurements were expected for the IT of the insertion device recording and for the Osstell and Penguin devices in the initial measurements. The CC implants were always more stable than the IH implant designs after adjusting for bone type and length. A higher stability was observed in dense bone than in soft bone for the same implant design and length. A higher stability was also observed for long implants versus short implants with the same bone type and implant design.

Generally, there was a significant positive correlation between the IT and the Osstell and Penguin device values in both the initial and the final measurements. A high and positive correlation was found between the Osstell and Penguin devices in the final measurements.

The IT and the initial measurements by the Penguin and Osstell devices in dense bone were not correlated, but in soft bone the correlations were positive and significant. The final measurements for the Osstell and Penguin devices and the IT values were positively and significantly correlated in both dense and soft bone. Implant length did not affect the correlation between the primary stability measurements by the different RFA devices. The correlation between the different implant design types of CC and IH revealed that in the CC design, the correlation between the RFA devices was maintained, while in the IH implant design the correlation was high and positive between the Osstell and Penguin devices only in the initial and final measurements. In the IH implant design, no correlation was observed between IT and the Osstell device or between IT and the Penguin device.

Although this is a laboratory model, it allows for a pure mechanical examination of the devices excluding any biological variance bias, such as for different bone types. This is a standardized model with constant conditions, independent of examiner or patient variability. The obvious disadvantage of this model is the absence of the biological impact of bone properties [19]. The mineral bone density of the posterior maxilla is 0.31 g/cm^3^ and that of the anterior maxilla is 0.55 g/cm^3^ [20]. The cortical thickness of the mandible is 2.22 mm, while the thickness in the maxilla is 1.49 mm [13]. Our block density was slightly softer and slightly denser in order to examine extreme cases. Polyurethane blocks still have the disadvantage of being a homogenous material that is dissimilar to real implant bed bone, which is neither entirely soft nor dense and can sometimes present heterogeneity in the bone quality, density, and elasticity [21].

The RFA measurements were performed when the implant was partially inserted, since it was impossible to insert the whole implant with the surgical motor and so a second, manual insertion with a ratchet was required. In order to correlate the IT values of the surgical motor with the RFA devices, measurements were performed at the point of partial implant insertion. Our findings are in accordance with the results of Turkyilmaz et al., who analyzed 30 Brånemark implants placed in the mandible and reported a Spearman correlation of 0.89 between the RFA and IT [22]. The similarity with our results may be explained by the placement of the implants in the uniform bone type of the edentulous mandible. Baldi et al. found a positive correlation between the IT and RFA values in a clinical multicenter study of 75 patients who had had conical implants inserted with knife edge threads. Although their rho coefficient was lower than in our correlation, it was still significant [23].

Becker and his group compared the Osstell and the Penguin devices with respect to immediate implant placement. The results of their study showed that there is a systematic bias between the devices, and in addition there is a low correlation between the instruments at the time of implant placement (correlation coefficient = 0.51) but a moderate correlation between them (0.71) at stage 1 [24].

A study that evaluated the correlation between the IT and primary stability of dental implants using different block bone densities found a strong and statistically significant correlation coefficient between the ISQ and IT. The values of both parameters were increased according to bone density [25].

In contrast to our findings, those of Acil and his group revealed no correlation between peak IT and RFA in self-cutting implants in a porcine bone model [26]. Degidi et al. clinically examined 4135 implants and found a low correlation between RFA and IT. They claimed that these two parameters are independent and represent different features of primary stability, since they represent different forces. The data showed that only the IT is influenced by bone density and that only the RFA is correlated to the length of the implants used [17]. RFA represents resistance to a bending force, while IT represents resistance to a shear force. The ISQ values for the Penguin and Osstell devices reflect that resistance to a perpendicular direction of the screw force, but the IT reflects the axial direction force [13].

In their systematic review aiming to understand the relationship between the implant stability measurements obtained by IT and RFA, Legas et al. concluded that these parameters are independent and incomparable methods for measuring primary stability and that the clinician should employ only one method of evaluation for each implant [18].

In our study, we used two commercially available implant designs which have different connections and different designs (the IH and CC); therefore, we could not isolate the effect of the connection from the macro-topography on the ISQ. In the future, we may perform a study with the same fixture macro-topography but with a different connection. Since it is well documented in the literature that surgical modification and different drilling protocols can influence primary implant stability [27,28,29], we used a constant drilling sequence and protocol in order to avoid any effects this may have had on the results.

## 5. Conclusions

In our model, there is a high correlation between the IT and ISQ from a purely mechanical perspective.Bone type does not affect the high positive correlation between the initial and final Osstell and Penguin device measurements.Implant length does not affect the correlation between the primary stability measurements attained by the different RFA devices.Implant design affects the correlation between the IT and ISQ, which is maintained in the CC design. However, in the IH design it was maintained only between the RFA devices.

## Figures and Tables

**Figure 1 materials-14-05537-f001:**
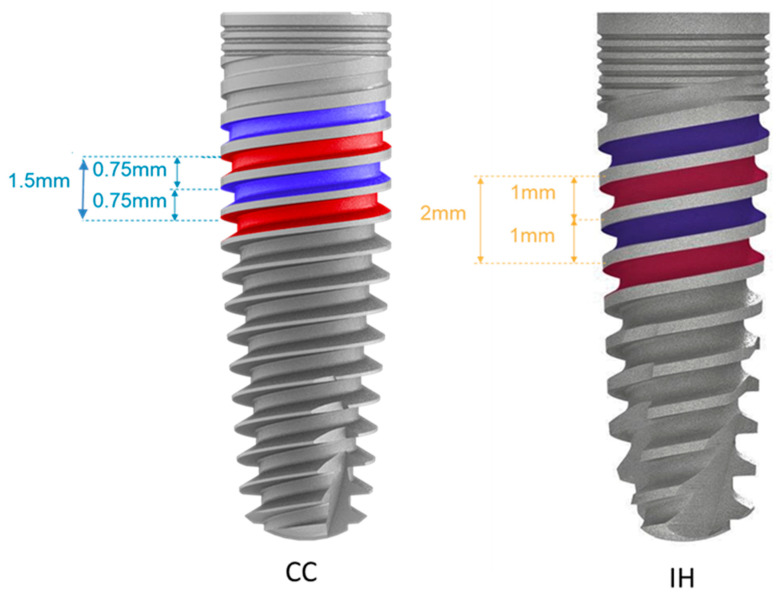
Implant design MIS. CC = Conical connection, IH = internal hexagon.

**Table 1 materials-14-05537-t001:** Experimental group design and sample size.

Implant Type, Diameter/Length	Soft Bone (4 Blocks)	Dense Bone (4 Blocks)	Total Implants
Design IH Ø 3.75/13 mm	12	12	24
Design CC Ø 3.75/8 mm	12	12	24
Design IH Ø 3.75/13 mm	12	12	24
Design CC Ø 3.75/8 mm	12	12	24
Total	48	48	96

**Table 2 materials-14-05537-t002:** The mean values of the three measurements for each device for all tested groups and the standard deviation.

Device	N	Mean	SD
IT (N/cm)	93	41.70	5.44
Osstell initial	93	56.87	5.42
Penguin initial	93	55.07	5.63
Osstell final	93	65.08	7.18
Penguin final	93	63.95	8.47

**Table 3 materials-14-05537-t003:** The mean values and standard deviations of implant stability for the three measurements by each device for each group with similar parameters and percentages of inserted implant length: (N = 12, 11) IT = insertion torque, CC = conical implant, and IH = implant.

Bone Type	Implant Length (mm)	Implant Type	Peak IT (N/cm)	Inserted Length %	Osstell Initial	Penguin Initial	Osstell Final	Penguin Final
Soft	13	CC	43 ± 3.13	88.81	62.21 ± 4.06	60.60 ± 4.61	67.33 ± 3.13	66.48 ± 2.09
Soft	13	HI	36.66 ± 4.14	86.19	51.18 ± 4.08	49.24 ± 3.69	57.54 ± 3.52	56.12 ± 4.14
Soft	8	CC	42.25 ± 2.14	85.68	56.83 ± 3.37	55.61 ± 4.74	62.66 ± 1.96	60.91 ± 2.65
Soft	8	HI	39.41 ± 4.12	87.14	48.97 ± 4.01	47.08 ± 2.99	53.61 ± 3.78	51.58 ± 3.38
Dense	13	CC	46.66 ± 3.20	61.54	59.55 ± 2.33	57.83 ± 2.64	73.38 ± 2.11	74.47 ± 2.35
Dense	13	HI	36.91 ± 3.65	72.73	61.90 ± 3.49	60.59 ± 4.56	68.96 ± 4.20	67.33 ± 6.62
Dense	8	CC	47.41 ± 4.78	62.5	59.33 ± 1.81	57.61 ± 2.03	72.13 ± 2.51	72.86 ± 3.34
Dense	8	HI	40.83 ± 5.78	66.67	56.11 ± 3.54	53.52 ± 3.92	64.88 ± 4.09	61.72 ± 6.13

**Table 4 materials-14-05537-t004:** Correlation coefficient between the different device measurements using Spearman’s rho correlation coefficient for all the different tested parameters (IT = insertion torque). ** The correlation is significant at the 0.01 level (2-tailed).

		Correlation Coefficient	Significance (2-Tailed)
Peak IT	Osstell initial	0.498 **	3.26 × 10^−7^
Peak IT	Osstell final	0.494 **	4.7 × 10^−7^
Peak IT	Penguin initial	0.486 **	6.77 × 10^−7^
Peak IT	Penguin final	0.529 **	5.06 × 10^−8^
Osstell initial	Penguin initial	0.953 **	1.78 × 10^−49^
Osstell final	Penguin final	0.964 **	2.42 × 10^−54^

**Table 5 materials-14-05537-t005:** Correlation coefficient between the different device measurements using Spearman’s rho correlation coefficient comparing the effect of bone type on the correlation (IT = insertion torque). ** The correlation is significant at the 0.01 level (2-tailed).

		Dense Bone		Soft Bone	
		Correlation Coefficient	Significance (2-Tailed)	Correlation Coefficient	Significance (2-Tailed)
Peak IT	Osstell initial	0.274	0.060	0.711 **	4.502 × 10^−8^
Peak IT	Osstell final	0.465 **	0.001	0.637 **	2.521 × 10^−6^
Peak IT	Penguin initial	0.273	0.061	0.684 **	2.252 × 10^−7^
Peak IT	Penguin final	0.494 **	0.000	0.599 **	1.408 × 10^−5^
Osstell initial	Penguin initial	0.977 **	1.09 × 10^−32^	0.955 **	2.702 × 10^−24^
Osstell final	Penguin final	0.955 **	1.93 × 10^−25^	0.983 **	2.693 × 10^−33^

**Table 6 materials-14-05537-t006:** Correlation coefficient between the different device measurements using Spearman’s rho correlation coefficient comparing the effect of implant length on the correlation (IT = insertion torque). ** The correlation is significant at the 0.01 level (2-tailed).

		8 mm Implant Length	Significance (2-Tailed)	13 mm Implant Length	Significance (2-Tailed)
Peak IT	Osstell initial	0.737 **	2.255 × 10^−9^	0.390 **	0.0073
Peak IT	Osstell final	0.500 **	2.974 × 10^−4^	0.556 **	7.293 × 10^−5^
Peak IT	Penguin initial	0.715 **	1.148 × 10^−8^	0.413 **	0.0043
Peak IT	Penguin final	0.550 **	5.208 × 10^−5^	0.563 **	5.630 × 10^−5^
Osstell initial	Penguin initial	0.924 **	7.963 × 10^−21^	0.974 **	4.094 × 10^−30^
Osstell final	Penguin final	0.963 **	1.024 × 10^−27^	0.964 **	2.267 × 10^−26^

**Table 7 materials-14-05537-t007:** Correlation coefficient between the different device measurements using Spearman’s rho correlation coefficient comparing the effect of the implant types CC and IH on the correlation (IT = insertion torque). ** The correlation is significant at the 0.01 level (2-tailed).

		CC	Significance (2-Tailed)	IH	Significance (2-Tailed)
Peak IT	Osstell initial	0.468 **	0.001	0.112	0.453
Peak IT	Osstell final	0.602 **	7.592 × 10^−6^	−0.074	0.626
Peak IT	Penguin initial	0.385 **	0.0075	0.100	0.502
Peak IT	Penguin final	0.571 **	2.758 × 10^−5^	−0.064	0.668
Osstell initial	Penguin initial	0.876 **	7.310 × 10^−16^	0.967 **	0.000
Osstell final	Penguin final	0.986 **	1.447 × 10^−36^	0.937 **	0.000

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
