# Peer review of "Reliability and Correlation of Different Devices for the Evaluation of Primary Implant Stability: An In Vitro Study"

_materials, 2021, doi:10.3390/ma14195537_

Round 1

Reviewer 1 Report

The study of “Reliability and correlation of different torque devices: An In Vitro Study of the Osstell and Penguin Implant Stability Quotient versus the Implant Insertion Torque” used two RFA devices Osstell and Penguin to measure the primary implant stability (ISQ) as well as insertion torque (IT) to analyze the effect of bone type, implant length and implant design in the primary implant stabilities.

Overall, it is an interesting and clinical orientated study. However, there are some major concerns needed to be clarified.

  1. It is necessary to present the figures related to the sample (including implant placed in the sawbones model and the measure process of ISQ and IT.)
  2. What is the number of samples in each group?
  3. There are several types of Sawbones for simulating trabecular bone. Which one should be used? In addition, why not use the soft bone block of sawbones model with (closed cellular foam) trabecular structure which is more closed to the real type of human bone. If you use it instead, the result may be completely different.
  4. What is the definition of partial implant length insertion?
  5. The authors concluded that “Implant design affects the correlation between IT and ISQ..”. However, it looks like that not only the connection type is different, but the fixture designs (included thread type and pitch) are also different, how to distinguish which one really has an impact on ISQ, or is it impossible to distinguish in this article?

Reviewer 2 Report

The manuscript entitled “Reliability and correlation of different torque devices: An In Vitro Study of the Osstell and Penguin Implant Stability Quotient versus the Implant Insertion Torque" submitted to Materials is an in vitro article on the correlation among many tools to analyze the primary implant stability.

This topic could be interesting for the readers, however it is not an innovative idea in the dental implantology scenario. I suggest many changes to improve the quality of manuscript to grab the reader’s attention.

Title:

I suggest to modify the manuscript title, in fact Osstell and Penguin are not torque devices.

I suggest to change the title focusing on primary implant stability.

Osstell and Penguin work on the analysis of resonance frequency.

Abstract:

Explain in a better way the topic of the manuscript and the null hypothesis.

In many parts the abstract seems confusing and not clear.

Introduction:
“Primary implant stability is affected by various factors, such as implant geometry, bone quality and quantity 36 (mainly cortical bone thickness) and the surgical drilling technique. [3]”

I suggest to include these recent studies [DOI: 10.1016/j.joms.2007.04.017 - DOI: 10.3390/dj8010021 - DOI: 10.3290/j.qi.a39745]

“Penguin (Integration Diagnostics, Sweden) is a similar device, with the exception that it 50 is electronic, as opposed to Osstell, which is a magnetic detection device. [7] Rittel and 51 his colleague, in their laboratory study, claimed that the sensitivity of RFA to changes in 52 the mechanical properties of periprosthetic tissue seems relatively weak and that RFA 53 might be more adapted to bone-healing estimations. [8]”

I suggest to evidence the presence of a conversion scale of values between Osstell and Penguin.

“Our hypothesis was that the different RFA devices Penguin and Osstell could provide 61 similar assessments of primary implant stability in similar clinical conditions: we ex- 62 pected to find a positive correlation between IT measurements conducted by the Pen- 63 guin and Osstell devices.”

Please, reformulate in a better way this sentence.

Methods

I don’t understand the reason to perform a RFA when the implant was partial inserted.

Explain clearly this part in the discussion part.

Discussion                 

“Primary stability is an essential parameter for immediate implant installation and early 202 loading…”

I suggest to report again the factors that could influence primary implant stability adding references about very recent surgical techniques such as bone compaction and osseodensification to improve primary implant stability.

“Although this model is a laboratory model, it allows for a pure mechanical examination 228 of the devices excluding any biological variance bias, such as different bone types. This 229 is a standardized model with constant conditions, independent of examiner or patient 230 variability. The obvious disadvantage of this model is the lack of the biological impact of 231 bone properties. [11] The mineral bone density of the posterior maxilla is 0.31 g/cm3, and 232 that of the anterior maxilla is 0.55 g/cm3 [12]. The cortical thickness of the mandible is 233 2.22 mm, while the thickness in the maxilla is 1.49 mm. [13] Our block density was 234 slightly softer and slightly denser, in order to examine extreme cases..”

I suggest to add other limitation of the study due the synthetic sample used.
I suggest to add the disadvantages of polyurethane solid rigid block.

The excessive homogeneity of the block cause the impossibility of making a sample as close as possible to the mandibular or maxillary human bone. In fact, there are very few clinical cases where the implant site is entirely of type D2 or D4, but presents bone heterogeneity along the implant bed. Another factor is the elasticity due to the displacement of the bone trabeculae during implant placement.

The use of polyurethane blocks does not affect the quality of the manuscript, however I feel it necessary to list the disadvantages of using this type of sample.

Conclusion

“Clinical relevance: 276

    In general, since an RFA device is not always available to the clinician and since 277 there is a positive correlation between the RFA reading and the IT registered on the in- 278 sertion device, one can rely on the latter for initial implant stability. 279
    The CC implant design has a higher initial stability on both initial and final RFA 280 readings, and both readings are correlated with the IT. On the other hand, in the IH de- 281 sign, the RFA and IT are not always correlated. This means that in the absence of an RFA 282 device, the clinician may confidently rely on the IT reading and make a decision for im- 283 mediate implant loading only when the CC design is inserted.”

I suggest to delete this part because this is an in vitro study on synthetic sample.

Improve the conclusion part and structure it in a better form.

Add an Abbreviation part at the end of the manuscript

Round 2

Reviewer 1 Report

No more comments. The authors have answered all the concerns and edited the manuscript.

Author Response

No Comments

Reviewer 2 Report

Authors followed the reviewer's suggestions.

I recommend the authors to check the order of references.
(Check ref. 5 and 6)

"The correlation between IT and RFA has been investigated in numerous studies but is still unclear; some authors claim that the two parameters are in a direct relationship [910], while others demonstrate no statistically significant correlations between the two."

I suggest to add recent literature about the analysis of IT and ISQ performed during an in vitro study [PMID: 32098046]

I suggest to add a part about the analysis of Insertion and removal torque in the introduction section.

I strongly suggest to the authors to refer to recent literature.
